# Insights into Thai and Foreign Hemp Seed Oil and Extracts’ GC/MS Data Re-Analysis Through Learning Algorithms and Anti-Aging Properties

**DOI:** 10.3390/foods14213739

**Published:** 2025-10-31

**Authors:** Suthinee Sangkanu, Thanet Pitakbut, Sathianpong Phoopha, Jiraporn Khanansuk, Kasemsiri Chandarajoti, Sukanya Dej-adisai

**Affiliations:** 1Department of Pharmacognosy and Pharmaceutical Botany, Faculty of Pharmaceutical Sciences, Prince of Songkla University, Hat Yai 90112, Songkhla, Thailand; suthinee.s@psu.ac.th (S.S.); jiraporn.kha@psu.ac.th (J.K.); 2Pharmaceutical Biology, Department of Biology, Friedrich-Alexander-Universität Erlangen-Nürnberg (FAU), 91058 Erlangen, Germany; thanet.pitakbut@fau.de; 3Department of Evolution and Population Biology, Institute for Biodiversity and Ecosystem Dynamics, University of Amsterdam, 1098 XH Amsterdam, The Netherlands; 4Traditional Thai Medical Research and Innovation Center, Faculty of Traditional Thai Medicine, Prince of Songkla University, Hat Yai 90112, Songkhla, Thailand; sathianpong.p@psu.ac.th; 5Department of Pharmaceutical Chemistry, Faculty of Pharmaceutical Sciences, Prince of Songkla University, Hat Yai 90112, Songkhla, Thailand; kasemsiri.c@psu.ac.th

**Keywords:** hemp seed, anti-aging properties, anti-elastase activity, HCA, PCA, PLS-DA, molecular docking, synergistic effect

## Abstract

This study successfully established a novel discriminative model that distinguishes between Thai and foreign hemp seed extracts based on gas chromatography/mass spectrometry (GC/MS) metabolic profiling combined with machine learning algorithms such as hierarchy clustering analysis (HCA), principal component analysis (PCA), and partial least square-discriminant analysis (PLS-DA). The findings highlighted significant metabolic features, such as vitamin E, clionasterol, and linoleic acid, related with anti-aging properties via elastase inhibition. Our biological validation experiment revealed that the individual compound at 2 mg/mL exhibited a moderate elastase inhibitory activity, 40.97 ± 1.80% inhibition (n = 3). However, a binary combination among these metabolites at 1 mg/mL of each compound demonstrated a synergistic effect against elastase activities up to 89.76 ± 1.20% inhibition (n = 3), showing 119% improvement. Molecular docking experiments aligned with biological results, showing strong binding affinities and enhanced inhibitory effects in all combinations. This integrated approach provided insights into the bioactive compounds responsible for anti-aging effects and established a dependable framework for quality control and standardization of hemp seed-based skincare products. Additionally, the developed models enable effective discrimination between Thai and foreign strains, which is valuable for sourcing and product consistency. Overall, this research advances our understanding of hemp seed phytochemicals and their functional potential, paving the way for optimized natural anti-aging formulations and targeted functional foods.

## 1. Introduction

*Cannabis sativa* L., or hemp, is a plant belonging to the genus *Cannabis* and family Cannabaceae, like *C. indica*. Cannabis is divided into two different categories—hemp and marijuana—based on its intended use. Due to its intoxicating properties, marijuana is mostly used recreationally, although it may also have therapeutic uses. On the other hand, hemp is significant due to its medical properties, fiber, and seed, all of which are used to produce a wide range of products [1]. Hemp fibers are proven to be strong, so they are utilized to make paper, sails, and clothes [2]. The hemp plant has several parts that are useful as food and as ingredients for supplements. While the leaves, sprouts, and hemp flowers can be eaten raw in salads and drinks, the seeds are the most consumed part of the plant. They offer 500–600 kcal/100 g. Additionally, hemp seeds contain one-fourth protein, one-fourth carbohydrate, and one-third fat, with some significant variations among different genotypes [3]. Hemp oil has essential fatty acids (EFAs), specifically, two polyunsaturated fatty acids (PUFAs), linoleic acid (LA, omega-6) and α-linolenic acid (ALA, omega-3). In addition, hempseed oil contains γ-linolenic (GLA) and stearidonic acid (SDA), which can only be found in a few plant families. Moreover, the tocopherols, phytosterols, and other bioactive compounds found in hemp oil help to prevent oxidation, inhibit free radical activity, and enhance its nutritional value and overall sensory qualities [4,5].

Integrating hemp actives into cosmeceuticals offers sustainable and natural substitutes for conventional skincare products with a variety of advantages, such as moisturizing and anti-aging effects [6]. The most researched phytocannabinoids, cannabidiol (CBD) and ∆-9-tetrahydrocannabinol (THC), both markedly reduced senescence as indicated by β-galactosidase activity while promoting cell proliferation in a dose-dependent manner. In addition, THC and CBD both markedly enhanced wound healing in both healthy and stress-induced premature senescence fibroblasts using a scratch assay [7]. By activating the PI3K/Akt pathway, CBD inhibited collagen breakdown and showed cytoprotective effects against UV-induced fibroblast alterations [8]. Besides phytocannabinoids, hemp seed extract also features a notable profile of fatty acids. The essential fatty acids included in hemp seed oil influence psoriasis, acne, and atopic dermatitis. α-linolenic acid and linoleic acid both diminish hyperpigmentation and UV damage [9]. Furthermore, hemp seed oil is a dry oil that does not cause acne and does not leave the skin feeling oily and sticky. Consequently, hemp seed oil has been used to create stable emulsions and long-lasting moisturizing patches for sunscreen cosmetics [10]. Hemp seed oil extraction yields seed paste residue that can be repurposed as a dermo-cosmetic agent utilizing ecologically friendly techniques like supercritical fluid extraction or extraction with ultrasonic support [11]. They demonstrated over 80% inhibition of the collagenase enzyme [11]. Cannabis acid derivatives, lignamides, amides, and a phenolic acid are among several bioactive substances present in these pastes.

Hemp cultivation began in China around 2700 BC. After that, cultivation moved throughout Asia [12] and around the world with domesticated varieties. Different types of hemp with distinct qualities have been created for various purposes during the domestication process [13]. In the same way, hemp seed metabolites may also be impacted by climatic and agronomic factors that affect reproduction. According to a metabolomic study, two indigenous hemp seed breeding variants in India have varying amounts of 236 metabolites [14]. In our previous study, the authors reported metabolic fingerprints of hemp seed oil and hemp seed extracts of two Thai and two foreign cultivars using GC/MS. Sixty-one metabolites were identified, and they demonstrated possible effects like antibacterial and anti-α-glucosidase activities [15]. With this complex original data, observing a distinctive relationship between metabolic features and hemp cultivar extract is impossible due to data complexity (high variance), which is a limitation. Supervised and unsupervised learning algorithms, like HCA, PLS-DA, and PCA, have been continuously reported to gain more insightful information from complex (high variance) GC/MS data by reducing data complexity and improving data visualization and classification.

The aim of this study is to differentiate between Thai and foreign hemp seed varieties and evaluate its anti-aging bioactivity using a combination of biological and chemometric methods. Therefore, the authors aim to re-analyze our previous GC/MS metabolomic fingerprints between Thai and foreign cultivars via these learning algorithms (HCA, PLS-DA, and PCA). Finally, the researchers analyzed the variations in active compounds present in hemp seeds to evaluate their anti-aging properties through the application of two enzymes, elastase and tyrosinase, alongside their anti-inflammatory effects, because elastase and tyrosinase are integral to skin elasticity and pigmentation. These findings support the further exploration of hemp seed-derived compounds as effective, natural ingredients in cosmetic formulations aimed at combating the signs of skin aging. This approach enabled a comparative assessment of the efficacy of hemp seeds derived from various strains, facilitating the identification of differing active ingredients. The findings from this study can inform the exploration and development of hemp strains that may be more advantageous for future applications.

## 2. Materials and Methods

### 2.1. Sample Preparation

In this study, we focused on examining variations in chemical composition among four different hemp seed extracts, which may lead to differing capabilities in inhibiting the aging enzyme. We also assessed the effectiveness of all extracts on the increased synthesis of nitric oxide (NO) in reaction to lipopolysaccharide (LPS) and cell survival. Four hemp seed samples were categorized into two groups. The first group included two samples from a Thai strain, specifically Hemp Seed-Thai-1 (HS-TH-1) (collection date 26 December 2022) and Hemp Seed-Thai-2 (HS-TH-2) (collection date 5 January 2023). The second group comprised two samples from a foreign strain, namely, Hemp Seed-Foreign-1 (HS-FS-1) (collection date 25 November 2022) and Hemp Seed-Foreign-2 (HS-FS-2) (collection date 26 December 2022). All samples were stored at 4 °C after they were received from the greenhouse. They were cleaned with water and dried at 40 °C in an incubator before extraction to obtain oil, ethanol, and hexane extracts, as reported in a previous study [15]. Briefly, approximately 2 kg of each sample were processed using an oil-press extractor, ensuring that the temperature remained below 40 °C to produce hemp seed oil, designated as HS-TH-1-O, HS-TH-2-O, HS-FS-1-O, and HS-FS-2-O. The residue was then separated for extraction with 80% ethanol and hexane. The materials were soaked in these solvents at room temperature for three days, and this extraction cycle was repeated three times. The resulting extracts were filtered, and the solvents were removed with a rotary evaporator (Heidolph, Schwabach, Germany). This produced four samples of crude extract in 80% ethanol (HS-TH-1-M-E, HS-TH-2-M-E, HS-FS-1-M-E, and HS-FS-2-M-E) and four samples of extract in hexane (HS-TH-1-M-H, HS-TH-2-M-H, HS-FS-1-M-H, and HS-FS-2-M-H). The yield of oils and extracts from hemp seed was shown in a previous report [15].

### 2.2. Data Preparation, Visualization, and Learning Algorithms

#### 2.2.1. Data Preparation and Visualization

The GC/MS data were manually extracted from the authors’ previous report [15] and kept in the proper format, XLSX and CSV files, for further evaluation. Three datasets were used in this study. The first dataset was the original data extracted from the previous report [15], with sixty-one metabolic features. The second dataset was the 10% GC/MS relative abundance cutoff data with thirteen major metabolic features, which aimed to minimize the noise signals from minor features to improve statistical analysis and learning algorithm performance. Finally, the third dataset was the four metabolic features, which were further selected after the thirteen-feature analysis; see Section 3.1. The last dataset served the same purpose as the second dataset, aiming to further reduce noise signals and increase model performance.

#### 2.2.2. Statistical Analysis and Learning Algorithms

Following a previous report [16], all data and statistical analysis were conducted using Python (version 3.13.3) via Jupyter Notebook (version 6.4.12) (Project Jupyter, https://jupyter.org/, accessed on 18 September 2025). With default parameter settings, the authors utilized the Seaborn (version 0.13.2) library for HCA analysis and the Scikit learn library (version 1.7.0) for PCA and PLS-DA analysis. Basic Python packages for mathematics and statistics, like NumPy (version 2.2.5) and Pandas (version 2.2.3), were used across the analysis, including determining R^2^ (coefficient of determination) and Q^2^ (predictability parameter) values and the permutation test with one thousand runs for model validation.

In PLS-DA analysis, the R^2^ value was calculated using Equations (1)–(3). On the other hand, for Q^2^ value calculation, a 5-fold cross-validation method was applied, and Equations (4)–(6) were used. Finally, the *p*-value of the permutation test was calculated using Equation (7).R^2^ = 1 − Sum of Squares Errors (SSE)/Total Sum of Squares (SST)(1)(2)$$\mathrm{SSM}=\sum {\left(y_{i}-\hat{y_{i}}\right)}^{2}$$(3)$$\mathrm{SST}=\sum {\left(y_{i}-\bar{y_{i}}\right)}^{2}$$Q^2^ = 1 − Predictive Errors Sum of Squares (PRES)/Total Variance (TSS)(4)(5)$$\mathrm{PRES}=\sum_{k=1}^{5}{\left(y_{i}-\hat{y_{i}}\right)}^{2}$$(6)$$\mathrm{TSS}=\sum_{k=1}^{5}{\left(y_{i}-\bar{y_{i}}\right)}^{2}$$*p*-value = Times in Permutation Q^2^ ≥ Observed Q^2/^Permutation Runs(7)
where $y_{i}$ $\hat{y_{i}}$, and $\bar{y}$ represent the observed, predicted, and mean value, and k represents cross-validation folds.

### 2.3. Anti-Elastase Assay

Anti-elastase activities of hemp seed extracts were investigated by a spectrophotometric assay with modification [17]. Briefly, 25 µL of 10 mg/mL of extract solution was mixed with 25 µL of 0.3 unit/mL elastase enzyme (CAS no. 39445-21-1) in 0.2 mM Tricine-HCl buffer (pH 8.0) and then incubated at room temperature for 15 min. Then, the solution of 1.6 mM N-Succinyl-Ala-Ala-Ala-*p*-nitroanilide was added. The mixture’s absorbance was immediately measured at 410 nm and continuously measured for 30 min using a microplate reader Thermo LUX PH5404 (Thermo Scientific, Waltham, MA, USA)). Dimethyl sulfoxide (DMSO) served as the negative control, while gallic acid was employed as the positive control. The experiments were performed in triplicate. The result was reported as percentage of elastase inhibition which was calculated using the following equation (Equation (8)):% Inhibition = [(Acontrol − A sample)/A control] × 100(8)

For the combination test, the experimental design reduced the concentration of pure compounds by half to investigate potential synergistic interactions and to inform future studies.

### 2.4. Anti-Tyrosinase Activity

Mushroom tyrosinase (CAS No. T3824-50KU) and L-DOPA were purchased from Sigma-Aldrich, MO, USA. Tyrosinase activity was determined by spectrophotometry, as described by Dej-adisai et al. [18]. In total, 140 µL of phosphate buffer pH 6.8, 20 µL of sample solution (200 µg/mL), and 20 µL of tyrosinase enzyme solution (203.3 unit/mL) were mixed in a 96-well plate at 25 °C for 10 min. Then, 20 µL of L-Dopa (0.85 mM) was added, and optical density (OD) at 492 nm was detected. After incubation at 25 °C for 20 min, the increase in absorption at 492 nm was monitored. DMSO was used as a negative control. Kojic acid and water extract of *A. lacucha* wood were used as positive controls. The percentage inhibition of tyrosinase reaction was calculated according to Equation (8).

### 2.5. Assay of Nitric Oxide (NO) Production

#### 2.5.1. Cell Culture

Murine macrophage RAW 264.7 cells were cultured in Dulbecco’s modified Eagle’s medium (DMEM) containing 10% fetal bovine serum, 2 mM glutamine, 1 mM pyruvate, penicillin (100 U/mL), and streptomycin (10 µg/mL). The cells were cultured at 37 °C in a humidified incubator with an atmosphere of 5% CO_2_. RAW 264.7 cells were seeded at 8 × 10^5^ cells/mL in 24-well plates and activated by incubation in medium containing LPS (1 μg/mL) in the presence of various concentrations of test compounds dissolved in DMSO. The supernatants were collected as sources of secreted NO.

#### 2.5.2. Cellular Viability Test

RAW 264.7 cells were cultured in 96-well plates for 18 h, followed by treatment with LPS (1 μg/mL) in the presence of plant extracts at concentrations of 400 μg/mL. After a 24 h incubation, 3-(4,5-dimethylthiazol-2-yl)-2,5-diphenyltetrazolium bromide (MTT) was added to the medium for 4 h. Finally, the supernatant was removed, and the formazan crystals were dissolved in DMSO. Absorbance was measured at 540 nm. The percentage of dead cells was determined relative to the control group.

#### 2.5.3. Nitric Oxide Assay

The determination of nitric oxide (NO) production from macrophages through the measurement of nitrite (NO^2−^) concentration in culture supernatants is a common assay [19]. Samples (100 µL) of culture media were incubated with 150 µL of Griess reagent (1% sulfanilamide and 0.1% naphthyl ethylene diamine in 2.5% phosphoric acid solution) at room temperature for 10 min in a 96-well microplate. Absorbance at 540 nm was read using a microplate reader Thermo LUX PH5404 (Thermo Scientific, MA, USA). Standard calibration curves were prepared using sodium nitrite as a standard.

### 2.6. Statistics

The data were evaluated using one-way variance analysis (ANOVA) for mean differences between different extracts, followed by a *T*-test. The results were expressed as the mean ± SD of three determinations. The threshold for statistical significance was set at *p* < 0.05.

### 2.7. Molecular Docking Simulation

We conducted a molecular docking simulation with slight adaptation following the authors’ previous reports [15,20]. In brief, the porcine elastase enzymatic crystal structure and its dipeptide inhibitor (PDB ID: 2EST) were obtained from the PDB database [21]. The authors used the UCSF Chimera program [22] to remove water and ion molecules. For docking experimental validation, the native inhibitor was extracted from its original position at the active site of the elastase enzyme, which was later redocked back to its original position. The authors utilized Open Babel software (version 2.3) [23] to generate the appropriate file format for the docking simulations, followed by simulation using AutoDock Vina version 1.2.5 [24].

For the docking experimental setup, the active (catalytic) site and its neighboring area were combined and used as a docking site. The coordinates and size of this docking site are shown below: X = 11.5, Y = 48.2, and Z = 2.4, and 30 Å × 25 Å × 20 Å. The authors used default values for nearly all docking parameters except exhaustiveness, adjusted to 36, and the number of docking outputs, set to 20. This experimental setup was used for both single and multiple ligand docking.

## 3. Results

### 3.1. Gas Chromatography/Mass Spectrometry (GC/MS) Data Visualization and Learning Algorithms

In this study, we re-analyzed our previously reported GC/MS data of two Thai (HS-TH-1 and HS-TH-2) and two foreign (HS-FS-1 and HS-FS-2) hemp seed cultivars using learning algorithms: HCA, PCA, and PLS-DA. Each hemp seed cultivar was extracted three times and named hemp seed oil (O), hemp seed meat hexane (M-H), and hemp seed meat ethanolic (M-E) extracts. First, the authors analyzed the original GC/MS data, which included sixty-one metabolic features from all twelve samples using HCA, PCA, and PLS-DA analyses. However, we could not observe a distinct pattern from the original GC/MS data via HCA and PCA analyses (Appendix A, Appendix A). On the other hand, the PLS-DA analysis was able to differentiate Thai from foreign cultivars with a high R^2^ value of 0.9850, indicating high model fit, yet the predictive power of the PLS-DA model was in a negative value, Q^2^ = −0.7665, showing a low predictability of the model (Appendix A, Appendix A). Therefore, the authors introduced a 10% GC/MS relative abundance cutoff threshold to eliminate noise signals from minor chemical metabolic feature influences for better analysis. The authors selected a 10% threshold criterion following a previous report by Santajit and team, considering any metabolite with ≥10% relative abundance as a major component [25].

Thirteen metabolic features remained after introducing a 10% cutoff, and the HCA analysis improved. As demonstrated in Figure 1, an observable correlation between the hemp seed cultivar extracts and the remaining metabolic features was possible. The current HCA analysis suggests that linoleic acid (omega-6) and clionasterol (γ-sitosterol) are the most significant metabolic features to distinguish between hemp seed cultivars.

Then, the authors performed PCA analysis on the 10% cutoff GC/MS data with the thirteen remaining metabolic features. The result showed an unclear differentiation with a considerable overlap between the two cultivars’ hemp seed extracts from the PCA analysis, with 45% of the cumulative variance (Figure 2, left; PC1 27% and PC2 19%). The authors used Pearson’s correlation matrix for further metabolic feature selection and optimization (Appendix A, Appendix A). As a result, four metabolic features (linoleic acid, clionasterol, linoleic acid ethyl ester, and oleic acid ethyl ester) were selected from six correlation clusters (Appendix A, Appendix A). Then, the PCA was re-analyzed based on these four newly selected metabolic features. Even though the re-analysis of PCA showed a significant enhancement in capturing the cumulative variances, increasing up to 87% (PC1 54% and PC2 33%), a slight improvement was observed in the differentiative pattern, with less overlap in data, between hemp cultivar extracts (Figure 2, right).

Then, a PLS-DA analysis was performed on both thirteen and four metabolic features (Figure 3). The PLS-DA analysis showed a more precise, distinctive pattern from thirteen metabolic features (Figure 3A) than four features (Figure 3B), with better performance parameters on both R^2^ and Q^2^ values. While the PLS-DA analysis from four selected metabolic features only demonstrated a relatively low R^2^ value of 0.5583, with a moderate Q^2^ value of 0.3090 (Figure 3B), the analysis of the thirteen metabolic features showed a relatively high R^2^ value of 0.8827 and a moderate Q^2^ value of 0.3733 (Figure 3A).

Permutation testing with a thousand runs was conducted to validate the predictability of the thirteen-metabolic-feature PLS-DA analysis (Figure 3C). The *p*-values from the permutation test were less than 0.05, indicating that the Q^2^ values or the model predictability were not detected by chance. The PLS-DA’s variable importance projection (VIP) scores of all thirteen metabolic features were above one, indicating an essential role in chemical distinction between the two hemp seed cultivar extracts (Figure 3D).

### 3.2. Elastase Inhibitory

Twelve extracts from hemp seeds were evaluated for their capacity to inhibit the elastase enzyme. The findings of this study are summarized in Table 1. All extracts, except for HS-FS-2-M-E, demonstrated considerable elastase inhibition (over 80%) when compared to the positive control, gallic acid, which showed an efficacy of 97.82%. Hemp seed oil and extracts from Thailand were rich in essential fatty acids, particularly linoleic acid, which is the main compound. In contrast, the primary components of hemp seed oil and extracts from foreign sources included clionasterol, linoleic acid, α,β-gluco-octonic acid lactone (HS-FS-2-M-E), and glyceryl-linoleate (HS-FS-1-M-E).

### 3.3. Anti-Tyrosinase Assay

Research investigating the inhibitory effects of hemp seed extract on the tyrosinase enzyme found that the ethanol extract resulted in a slight inhibition in enzyme activity at a concentration of 20 µg/mL (4.91–17.63%). However, neither the oil nor the hexane extracts had any impact on the tyrosinase enzyme (Table 2).

### 3.4. Biological Evaluation and Docking Simulation of the Top Three Metabolic Features

#### 3.4.1. Biological Validation

Based on the PCA and PLS-DA analysis findings, the top three key metabolic feature compounds, i.e., vitamin E, clionasterol, and linoleic acid, were selected for further biological evaluation. As a result, these compounds were evaluated for their anti-elastase activity. Each compound inhibited the elastase enzyme by around 40%. After a two-component combination of these three metabolites, the percentage inhibition against porcine elastase was synergistically improved, as shown in Table 3. The best percent improvement was observed from the combination of clionasterol and linoleic acid, showing 119% and 118% improvement, compared to the individual, which reached an 89% elastase inhibitory effect. At the same time, the second-best improvement was from the combination of clionasterol and vitamin E, with 67% and 63% improvement relative to each component activity, showing nearly 67% inhibition against elastase activity. Finally, the combination of linoleic acid and vitamin E showed the smallest percent improvement, around 26–29%, compared to the single evaluation, with a total of 58% elastase inhibition.

#### 3.4.2. Docking Simulation

The authors performed a molecular docking simulation to provide theoretical support for validating the earlier biological finding, the anti-elastase activity of the selected three metabolites and their combination, and proposing potential molecular interactions between bioactive molecules and amino acid residues on elastase. First, the authors validated the molecular docking experimental setup via the redocking method, comparing the redocked extracted native molecule to its original position. This method ensures that the experimental setup for docking simulation can reproduce the native ligand pose that came with the elastase crystal structure (PDB ID: 2EST). As a result, the redocking validation result exhibited a root-mean-square deviation (RMSD) value of 2.661 Å, passing the standard criterion of less than 3 Å. This ensured the reliability of the authors’ setup docking experiment. The result of the redocking validation is provided in the Appendix A (Appendix A).

Later, the authors performed six independent docking simulations of single and combined molecules from the top three selected metabolic features. Each experiment’s best docking pose was chosen based on the following selection criterion: similar chemophores (chemical structural features or functional groups) will share their proximity interaction pose and binding sites. As a result, six docking poses were selected. Figure 4A,C,E demonstrate the single docking result of clionasterol, linoleic acid, and vitamin E. In contrast, Figure 4B,D,F exhibit the combined docking result of clionasterol–linoleic acid, clionasterol–vitamin E, and linoleic acid–vitamin E. Finally, Figure 4G shows a regenerated molecular interaction of the native inhibitor that came with the crystal structure of the elastase enzyme obtained from the PDB database, as a reference.

The molecular docking result showed highly conserved molecular poses of a cyclic structure among the docking experiments, as presented in both clionasterol and vitamin E single-molecule docking (Figure 4A,E) and combined-molecule docking (Figure 4B,D,F). Also, it exhibited preserved hydrogen bonds with the R61 amino acid from clionasterol’s results, both single- and combined-molecule docking, and with the Q192 amino acid from vitamin E docking results (Figure 4 and Table 4). On the other hand, molecular docking indicated a significant variation for aliphatic structures from linoleic acid (Figure 4B,C,F) and the partial structure of vitamin E (Figure 4D–F). Yet molecular docking suggested a relatively conserved hydrogen bond across the docking results for the hydroxyl group of linoleic acid with the T41 amino acid residues, except for the linoleic acid–vitamin E combined docking simulation (interacting with H40 instead of T41; Figure 4F).

Finally, the docking scores for each experiment are listed in Table 4. A lower score indicates a stronger theoretical interaction between the chemical molecule and the amino acid residue on the elastase enzyme, suggesting potential biological inhibition. Similarly to the biological validation in the section above, the percentage improvement of each docking score is also calculated and shown in Table 4.

### 3.5. Nitric Oxide Inhibitory Activity

Hemp seed extracts showed an inhibitory effect on RAW 264.7 murine macrophage cells, which resulted in a reduction in the production of nitric oxide. The hemp seed oil, HS-FS-2-O caused a decrease in nitric oxide production, while HS-TH-1-O, HS-TH-2-O, and HS-FS-1-O did not have any effect at 400 µg/mL. Ethanol extracts of hemp seed at concentrations of 100, 200, and 400 µg/mL, including HS-TH-1-M-E, HS-TH-2-M-E, and HS-FS-1-M-E, successfully reduced nitric oxide levels, whereas HS-FS-2-M-E did not lead to any reduction. Hexane extracts displayed similar outcomes compared to ethanol extracts but were less effective in lowering nitric oxide levels (Figure 5). This inhibitory effect is not attributed to cytotoxicity, as evidenced by cell viability values ranging from 70 to 110%.

## 4. Discussion

Phytochemical analysis is an essential toolkit to explore the complex chemistry of medicinal plants and is necessary for the cultivation and standardization of botanical materials used in pharmaceutical references. Integrating machine learning with modern analytical techniques has shown a promising role in the quality assurance for herbal medicines, where accurate identification of plants is critical to producing safe pharmaceutical and medicinal products with reliable effectiveness. In this work, the authors re-analyzed the previously reported GC/MS dataset, aiming to distinguish the strains in which cultivar variations affect the chemical profile of hemp seed extracts via learning algorithms. When all GC/MS data were subjected to HCA, no significant metabolite distinctive pattern was observed between Thai and foreign hemp seed varieties. Therefore, a 10% relative metabolic abundance cutoff was introduced. Thirteen primary metabolites remained and were re-evaluated using HCA. This analysis distinctly separated the primary metabolites of Thai and foreign hemp seeds. The Thai variety was characterized by linoleic acid (omega-6), while clionasterol (γ-sitosterol) was the predominant metabolite in the foreign variety. However, when PCA, an unsupervised learning method, was applied, it revealed that the variation among the thirteen metabolic traits was not the dominant factor driving the differences in classification [26]. As a result, unsupervised learning methods like PCA did not produce a clear separation. Then, PLS-DA (supervised learning) was applied. The PLS-DA analysis of thirteen metabolic features exhibited a high R^2^ value of 0.8827 with a moderate Q^2^ value of 0.3733. It indicated a good model fit based on the R^2^ value of nearly 0.9 and a moderate model predictability based on the Q^2^ value above 0.3 [27]. The permutation test showed a *p*-value less than 0.05, indicating non-randomness in the model’s predictability. In conclusion, based on the provided model parameters, the PLS-DA analysis was able to capture a discriminative pattern between Thai and foreign cultivars using the current 10% relative abundance cutoff GC/MS data, with thirteen metabolic features. However, external data has not yet been validated in this analysis, which limits its generalizability and robustness. Therefore, more information is needed to assess how the current PLS-DA performs under real-world conditions.

The various chemical components present in plants lead to diverse biological effects [28]. Hemp seed oil offers adequate sun protection, repair properties, anti-allergy benefits, and anti-aging effects, making it a valuable raw material for skincare product formulation [29]. Consequently, this study on hemp seed extracts concentrated on their potential in cosmeceuticals, whose research and development focuses on tackling the primary signs of aging. Various plant metabolites have been identified for their ability to modulate the activity of enzymes involved in the aging process. Notably, elastase and tyrosinase are significant enzymatic targets of interest in cosmetics [30]. Elastase is recognized for its role in causing rheumatoid arthritis, pulmonary emphysema, and various other chronic inflammatory conditions by breaking down human tissues. Additionally, it breaks down elastin, a vital protein essential for skin elasticity and restoration, which contributes to the formation of wrinkles [31,32]. In this investigation, wrinkles were identified as the initial focus of the study. All extracts, except for HS-FS-2-M-E, demonstrated considerable elastase inhibition (Table 1). Our results were consistent with the existing literature, where hemp extract demonstrated approximately 30% elastase inhibition at a concentration of 1 mg/mL [33]. However, these extracts showed no tyrosinase inhibitory activity at the specified doses, which was unrelated to the fatty acids in the hemp seed extracts. This finding is consistent with the study by Saleepochn et al. [34], which reported that the tyrosinase inhibitory activity of crude extracts from hemp seeds ranged from 11.1 to 29.2% when evaluated at a concentration of 0.75 mg/mL. Nonetheless, other reports have indicated that the ethyl acetate (EtOAc) fraction of hemp seed extracts, which contain phenethyl cinnamamides, demonstrated strong inhibitory effects on melanin production [35]. Therefore, the extraction process needs to be changed to enhance hemp seed extracts to inhibit the tyrosinase enzyme in the future.

Later, the top three key metabolic features, i.e., vitamin E, clionasterol, and linoleic acid from PLS-DA analysis, were subjected to biological evaluation. Our results demonstrated that each compound individually reduced elastase activity by approximately 40%. When two of these compounds were combined, their inhibitory effect on porcine elastase was synergistically enhanced (Table 3). Another study examined *G. bimaculatus* oil, which is high in linoleic acid (31.08 ± 0.00%), and found that it effectively inhibited elastase with a half-maximal inhibitory concentration (IC_50_) of 178.6 ± 62.0 µg/mL [36]. Linoleic acid (LA, C18:2n-6), also known as omega-6, is a polyunsaturated fatty acid (PUFA) present in vegetable and seed oils [37]. It had a notable effect on elastase activity at a low concentration of 10 µM [38]. Furthermore, it serves as a precursor to oxidized products, which can modulate pain and inflammatory responses [39]. Phytosterol components, clionasterol or γ-sitosterol, were found in hemp seed oil and extracts. They have been shown to possess biological activity, providing emollient and anti-aging benefits. They have been reported to encourage the production of hyaluronic acid, improve epidermal thickness, enhance skin elasticity, and decrease skin roughness. As a result, they are suitable for inclusion in anti-aging creams and sun protection products [40,41]. Vitamin E (α-tocopherol) aids in skin protection by diminishing inflammation and preventing excessive cell growth. Additionally, it enhances skin smoothness and helps retain moisture in the stratum corneum, thereby promoting overall skin health and appearance [42]. Its dietary sources are mainly vegetable oils and nuts such as wheat germ oil, olive oil, peanut oil, corn oil, almond oil, flaxseed oil, coconut oil, and sunflower seeds [43]. In our previous study, vitamin E was identified in hemp seed oil (HS-TH-1-O, HS-FS-1-O, and HS-FS-2-O) and hexane extract (HS-FS-2-M-H), which demonstrated an elastase inhibitory effect exceeding 80%. For HS-FS-2-M-E, the anti-elastase activity was measured at 64.84%. This extract mainly contained α,β-gluco-octonic acid lactone at 30.34%, with linoleic acid present as a minor component at 1.43% (Table 1). There are no reports indicating that α,β-gluco-octonic acid lactone possesses anti-elastase activity. Nevertheless, the inhibition of the enzyme observed in this study might be influenced by other components in the extract, such as polyphenols and cannabinoids [44].

The authors performed molecular docking to evaluate the theoretical possibility of molecular mechanistic interactions and binding poses of single and combined molecules of the selected three key metabolic features suggested by PLS-DA and in vitro analysis. The molecular docking result showed highly conserved molecular poses of single-molecule docking. Additionally, the docking scores obtained from the combined-molecule simulation followed the same trend as the biological validation, showing that combining clionasterol with linoleic acid and vitamin E exhibited more substantial inhibitory effects (−8.03 kcal/mol with 94% and 30% improvement and −10.03 kcal/mol with 91% and 63% improvement) than mixing linoleic acid and vitamin E (−6.56 kcal/mol with 58% and 25% improvement) (Figure 4 and Table 4). This work shows that re-analysis using algorithmic models permits more accurate analysis of the components in hemp seed extracts and decreases noise, which may lead to future research that can save both time and funds.

Numerous stresses, including oxidative damage, radiation exposure, and inflammatory conditions, cause cellular decline, which is a sign of aging [45]. Therefore, in this study, the authors also assessed the anti-inflammatory effects of hemp seed extract, which was shown to decrease nitric oxide production in RAW 264.7 murine macrophage cells. Hemp seed extracts showed no toxicity to cells at the tested concentrations (100, 200, and 400 µg/mL), and they also demonstrated strong anti-inflammatory effects. The extracts with notable activity include the oil extracts from foreign strains: HS-FS-2-O; ethanol and hexane extracts from Thai strains: HS-TH-1-M-E, HS-TH-2-M-E, HS-TH-1-M-H; and the ethanol extract from foreign strain seed: HS-FS-1-M-E (Figure 5). Similarly to the elastase inhibition discussed earlier, the authors investigated the relationship between metabolic features and nitric oxide inhibition using PLS-DA, based on a 10% GC/MS abundance threshold and classifications of nitric oxide inhibitory activity. We could not observe any meaningful correlation between the thirteen metabolic compounds and nitric oxide inhibition.

## 5. Conclusions

The potential benefits of hemp seed extracts in cosmeceutical applications, particularly for anti-aging and anti-inflammatory objectives, is highlighted by this extensive phytochemical and bioactivity investigation. The metabolic profiles of Thai and foreign hemp seed varieties could be distinguished using analytical techniques like GC/MS combined with multivariate statistical and machine learning approaches like HCA, PCA, and PLS-DA analysis. Vitamin E, clionasterol, and linoleic acid were shown as the top three discriminative features suggested by the PLS-DA analysis, a supervised learning method. These features contributed to high classification accuracy and demonstrated synergistic elastase inhibitory effects. These results have been confirmed by molecular docking, which showed that combinations of these compounds were able to improve enzyme inhibition theoretically. In conclusion, the extracts demonstrated potential biological activity at the tested concentrations without causing cytotoxicity, such as substantial anti-inflammatory effects through nitric oxide suppression. Therefore, this study showed that integrating phytochemical analysis with modern chemometric machine learning and molecular modeling methods not only provides valuable insights into the biologically beneficial properties of hemp seed derivatives but also indicates the potential to adopt this method as a quality control measurement to support the development of hemp seed extract skincare products as a natural-based anti-aging component.

## Figures and Tables

**Figure 1 foods-14-03739-f001:**
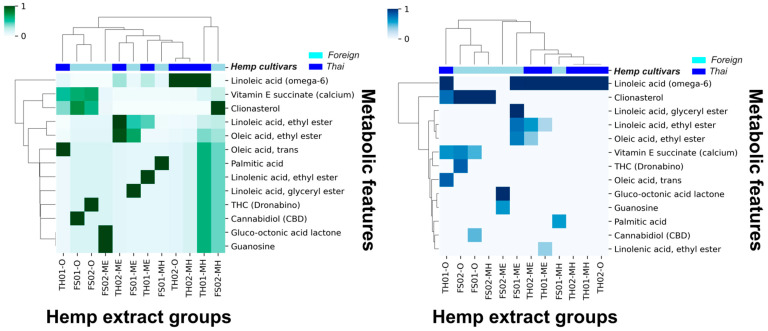
Hierarchical clustering analysis (HCA) of Thai and foreign hemp seed oil and extracts using z-score standardized 10% GC/MS relative abundance cutoff data (**left**, green) and using normal data (**right**, blue).

**Figure 2 foods-14-03739-f002:**
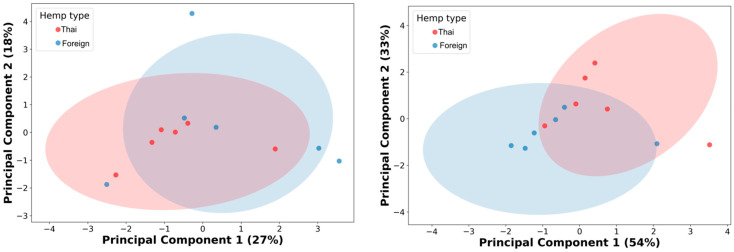
Principal component analysis (PCA) of 10% GC/MS relative abundance cutoff, thirteen metabolic features (**left**), and the four selected metabolic features (**right**). Red dots represent hemp extracts from Thai cultivars; blue dots indicate hemp extracts from foreign cultivars.

**Figure 3 foods-14-03739-f003:**
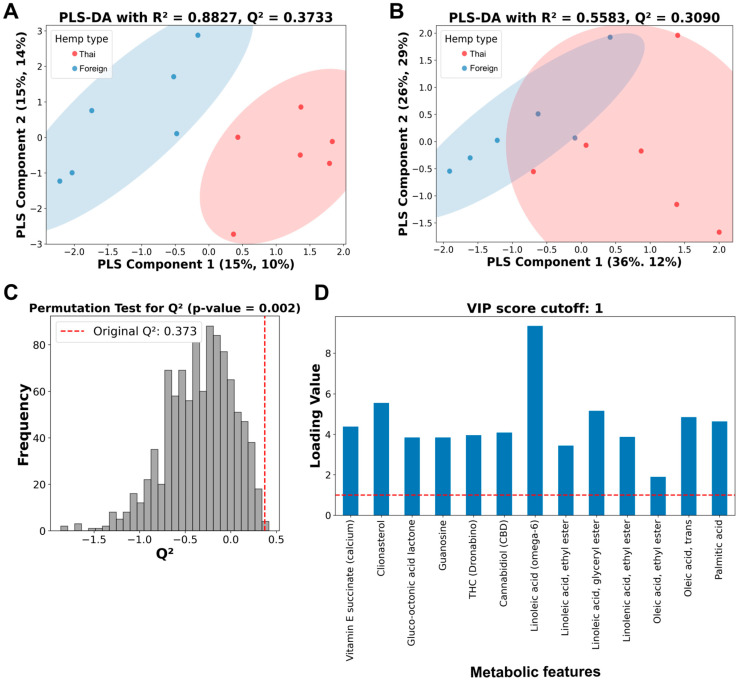
Partial least squares discriminant analysis (PLS-DA), permutation tests, and variable importance projection (VIP) score plots of selected metabolic features from Thai and foreign hemp cultivar extracts. (**A**) PLS-DA of 10% GC/MS relative cutoff with thirteen metabolic features. (**B**) PLS-DA of the four selected metabolic features. Red dots represent hemp extracts from Thai cultivars, while blue dots indicate hemp extracts from foreign cultivars. (**C**) Histogram plot of the permutation tests of the PLS-DA from A with the original Q^2^ value of 0.3733. (**D**) VIP score plot of thirteen metabolic features. Vertical and horizontal dashed lines indicate a position of Q^2^ value and a score of 1, indicating a significant threshold of metabolic features from the thirteen-metabolic-feature PLS-DA analysis, respectively.

**Figure 4 foods-14-03739-f004:**
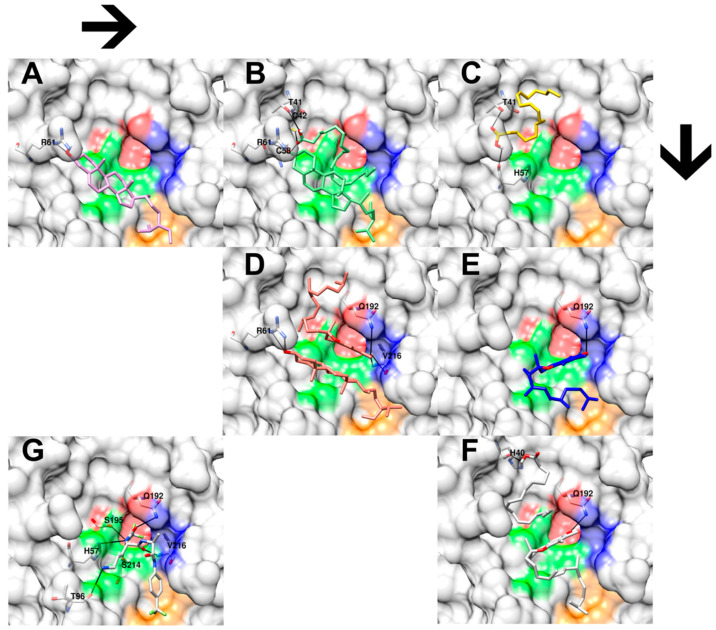
Three-dimensional close-up docking results of hemp seed oil extracts’ selected chemical features and their combinations on porcine elastase enzyme (PDB ID: 2EST). (**A**) shows the docking interaction of clionasterol (pink) and amino acid R6 (white). (**B**) shows the docking interaction of combined clionasterol and linoleic acid molecules (green) with amino acids T41, C42, C58, and R61 (white). (**C**) shows the docking interaction between linoleic acid (yellow) and amino acids T41 and H57 (white). (**D**) shows the docking interaction between combined clionasterol and vitamin E molecules (red) with amino acids R61, Q192, and V216 (white). (**E**) shows the interaction of vitamin E (blue) with amino acid Q192 (white). (**F**) shows the docking interaction of combined linoleic acid and vitamin E molecules with amino acids H40 and Q192 (white). (**G**) shows the molecular interactions between the native inhibitor from the crystal structure of porcine elastase and its amino acids H57, T96, Q192, S195, S214, and V216. The black lines indicate hydrogen bonds. Red, green, blue, and orange colors on the enzyme surface indicate the four porcine elastase catalytic subpockets, S1, S1’, S2’, and S3’, respectively.

**Figure 5 foods-14-03739-f005:**
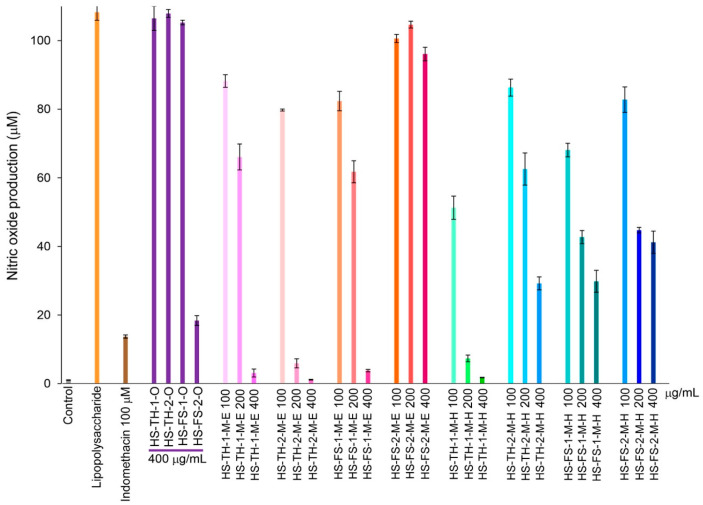
Changes in NO production rate over 24 h of incubation from hemp seed extracts. The error bars represent the standard deviation (SD) of the mean (n  =  3).

**Table 1 foods-14-03739-t001:** The elastase inhibitory activity of hemp seed oil and extract, along with the positive control.

Extract	% Elastase Inhibition (2 mg/mL)	% of Total		
		Linoleic Acid	Vitamin E	Clionasterol
HS-TH-1-O	89.48 ± 1.26	20.09 *	12.63	15.35
HS-TH-1-M-H	91.63 ± 2.05	22.93 *	-	6.66
HS-TH-1-M-E	86.18 ± 3.06	35.13 *	-	2.75
HS-TH-2-O	84.02 ± 2.07	86.53 *	-	1.53
HS-TH-2-M-H	93.17 ± 1.44	66.24 *	-	1.44
HS-TH-2-M-E	87.44 ± 0.39	34.08 *	-	1.55
HS-FS-1-O	92.62 ± 2.55	-	14.60	29.07 *
HS-FS-1-M-H	89.87 ± 3.50	17.63 *	-	5.67
HS-FS-1-M-E	85.00 ± 0.49	15.04	-	3.80
HS-FS-2-O	97.15 ± 0.89	-	15.10	22.32 *
HS-FS-2-M-H	93.05 ± 1.39	-	1.09	13.42 *
HS-FS-2-M-E	64.84 ± 1.67	1.43	-	-
Gallic acid	97.82 ± 0.24	Positive control		

% Elastase inhibitions are expressed as mean ± SD (n = 3) and compared with gallic acid as a positive control (*p* value < 0.05). * = % of the major component founded in hemp seed oil and extract as mentioned in previous publications. - = Not founded [15]. Abbreviation remarks: HS = Hemp seed; TH = Thai species; FS = Foreign species; O = Hemp seed oil; M = Hemp seed marc; H = Hexane extract; E = 80% Ethanol extract.

**Table 2 foods-14-03739-t002:** Evaluation of the potential activity of hemp seed oil and extracts on tyrosinase.

Extract *	% Inhibition ± SD (20 µg/mL)
HS-TH-1-O	−6.30 ± 2.13
HS-TH-1-M-H	−13.24 ± 4.55
HS-TH-1-M-E	11.53 ± 1.88
HS-TH-2-O	−9.83 ± 2.28
HS-TH-2-M-H	−3.26 ± 2.40
HS-TH-2-M-E	17.63 ± 2.59
HS-FS-1-O	−7.80 ± 8.02
HS-FS-1-M-H	−1.31 ± 5.57
HS-FS-1-M-E	4.91 ± 6.00
HS-FS-2-O	−3.14 ± 6.61
HS-FS-2-M-H	0.33 ± 8.52
HS-FS-2-M-E	12.59 ± 3.33
Kojic acid	89.04 ± 4.99
Water extract of *A. lacucha* wood	96.01 ± 3.40

* Abbreviation remarks: HS = Hemp seed; TH = Thai species; FS = Foreign species; O = Hemp seed oil; M = Hemp seed marc; H = Hexane extract; E = 80% Ethanol extract.

**Table 3 foods-14-03739-t003:** Elastase inhibitory activity of the selected compounds and their combination test.

	Clionasterol	Linoleic Acid	Vitamin E
Clionasterol	40.97 ± 1.80	89.76 ± 1.20 [119%, 118%]	66.94 ± 0.71 [67%, 63%]
Linoleic acid		41.15 ± 0.15	51.88 ± 0.22 [29%, 26%]
Vitamin E			40.08 ± 0.38

The individual compounds were evaluated at a final concentration of 2 mg/mL, while the combinations were assessed at a final concentration of 1 mg/mL for each compound. Inside the bracket [ ] is the percentage increase in elastase inhibitory activity of combined molecules compared to each single molecule.

**Table 4 foods-14-03739-t004:** Docking results regarding the selected three essential chemical features from hemp seed oil extract and porcine elastase enzyme (PDB ID: 2EST).

	PubChem ID	Docking Score (Kcal/Mol) [% Increase]	Type	Total	Residue (Bond)	Subunit	Catalytic Site
Single							
Clionasterol (γ-sitosterol)	457801	−6.17	H-bond	1	R61 (1)	1	No (near)
Linoleic Acid	5280450	4.14	H-bond	2	T41 (1) and H57 (1)	-	No (near)
Vitamin E	14985	−5.26	H-bond	1	Q192 (1)	S’1-S’2	Yes
Combination							
Clionasterol–Linoleic Acid	-	−8.03 [94%, 30%]	H-bond	4	T41 (1), C42 (1), C 58(1) and R61 (1)	-	No (near)
Clionasterol–Vitamin E	-	−10.03 [91%, 63%]	H-bond	5	R61 (2)		No (near)
					Q192 (1)	S1-S’1	Yes
					V216 (2)	S’1-S’2	Yes
Linoleic Acid–Vitamin E	-	−6.56 [58%, 25%]	H-bond	2	Q192 (1)	S’1-S’2	Yes
					H40 (1)	-	No (near)

Inside the bracket [ ] is the percentage increase in elastase activity of combined molecules compared to each single molecule.

## Data Availability

The original contributions presented in this study are included in the article and Appendix A. Further inquiries can be directed to the corresponding author.

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
