# Peer review of "Insights into Thai and Foreign Hemp Seed Oil and Extracts’ GC/MS Data Re-Analysis Through Learning Algorithms and Anti-Aging Properties"

_foods, 2025, doi:10.3390/foods14213739_

Round 1
Reviewer 1 Report
Comments and Suggestions for Authors
In the abstract, you report ~40% inhibition for single compounds and “up to 89%” for combinations—please add n, mean ± SD, exact concentrations, and the statistical test supporting “synergistic.”
When stating that PCA/HCA/PLS-DA are needed to distinguish strains, add a sentence clarifying prior limitations of your earlier dataset and why re-analysis was necessary
Please narrow the cosmeceutical scope: you discuss elastase, tyrosinase, and anti-inflammatory effects; explicitly frame elastase as the primary endpoint and the others as secondary.
Clarify harvest year, storage time, moisture content and lot identity for the four hemp seed batches, as these can shift metabolite profiles. Add these to Section 2.1
You extracted ~2 kg per sample and produced oil plus two marc extracts. Please state extraction yields (% w/w) for oil, hexane marc extract, and ethanolic marc extract, with mean ± SD
Specify GC–MS quantitation basis used later (e.g., normalized peak area %, internal standard, TIC area) so the “10% abundance cutoff” is interpretable and reproducible
Table 1: add n, exact p-values vs control, and effect sizes for elastase inhibition. Also clarify the asterisk meaning “major component” and ensure consistency with the prior publication.
Figure 1/2: Please increase font sizes, add units/axes labels, and make raw points visible (e.g., jitter over group ellipses). Supply high-resolution vector files.
Author Response
Reply to Reviewer 1
Comments and Suggestions for Authors
- In the abstract, you report ~40% inhibition for single compounds and “up to 89%” for combinations—please add n, mean ± SD, exact concentrations, and the statistical test supporting “synergistic.”
Response: Revised. Lines 27-30
- When stating that PCA/HCA/PLS-DA are needed to distinguish strains, add a sentence clarifying prior limitations of your earlier dataset and why re-analysis was necessary
Response: Revised. Lines 92-94.
- Please narrow the cosmeceutical scope: you discuss elastase, tyrosinase, and anti-inflammatory effects; explicitly frame elastase as the primary endpoint and the others as secondary.
Response: Revised. Lines 470-471.
- Clarify harvest year, storage time, moisture content and lot identity for the four hemp seed batches, as these can shift metabolite profiles. Add these to Section 2.1
Response: Revised. Lines 121-126.
- You extracted ~2 kg per sample and produced oil plus two marc extracts. Please state extraction yields (% w/w) for oil, hexane marc extract, and ethanolic marc extract, with mean ± SD
Response: Revised. All data were presented in our previous study, so we have added a reference for this data. Lines 135-136.
- Specify GC–MS quantitation basis used later (e.g., normalized peak area %, internal standard, TIC area) so the “10% abundance cutoff” is interpretable and reproducible
Response: In our previous report, the GC-MS data was shown in % relative abundance. Therefore, here, I would change the term from “10% abundance cutoff” to “10% relative abundance cutoff”. So, it is clear to the reviewers and the readers what GC-MS quantity was used in our current manuscript.
- Table 1: add n, exact p-values vs control, and effect sizes for elastase inhibition. Also clarify the asterisk meaning “major component” and ensure consistency with the prior publication.
Response: Revised. Lines 327-329.
- Figure 1/2: Please increase font sizes, add units/axes labels, and make raw points visible (e.g., jitter over group ellipses). Supply high-resolution vector files.
Response: The authors replaced a high-resolution figure as the reviewer suggested and increased the font size to improve readability. Furthermore, the authors introduced a jitter function into the analyses to make all data points visible (not overlapping) as suggested.
Thank you very much,
Best regards,
Authors
17.10.2025

Reviewer 2 Report
Comments and Suggestions for Authors
The study is well-designed and provides valuable insight into the chemical and biological differences between Thai and foreign hemp seed extracts. Combining GC-MS profiling with machine learning and biological assays is innovative and relevant for cosmeceutical applications. The results are solid, however, some minor clarifications and additions are needed to strengthen the manuscript.
- The title is informative but long. You may consider a shorter version, but not mandatory.
Abstract
- Avoid abbreviations (HCA, PCA, PLS-DA) before defining them.
- Add 1-2 lines introducing the project purpose e.g. “This study aimed to distinguish Thai and foreign hemp seed varieties and assess their anti-aging bioactivity through combined chemometric and biological approaches.”
- Emphasize the novelty and purpose more clearly.
Introduction
- Line 61-63: Add reference for CBD/THC reducing senescence.
- Line 67-68: Add reference for CBD’s effect on collagen breakdown via PI3K/Akt pathway.
- Line 72-75: Add citation for extraction methods (supercritical, ultrasonic).
- Line 75-76: Add reference for collagenase inhibition.
- Line 76-77: Clarify which compounds showed the inhibition, currently ambiguous.
- Line 96: Briefly explain the purpose of studying two enzymes (elastase and tyrosinase) in relation to aging and pigmentation.
Materials and Methods
- Line 121: Explain what “LPS-induced NO production” means.
- Line 160: Define “LPS” (lipopolysaccharide) at first mention.
- Add justification for the selected extract concentrations.
- Line 171: Avoid repeating previously defined abbreviations.
- Line 90 vs. 200: Ensure consistent use of “PCA” (Principal Component Analysis) everywhere, not “Partial Component Analysis”.
- Add statistical details (number of replicates, test used, significance/confidence level).
- Provide rationale for choosing the 10% GC-MS abundance cutoff.
Results & discussion
- Line 306-309: Verify figure numbering for single and combined docking images (A-F).
- Expand explanation of the docking and redocking validation process, briefly mention why redocking is performed (to ensure the docking setup can reproduce the native ligand pose) and explain docking score interpretation (more negative = stronger binding).
- Line 393: Move a short summary of this finding (hemp extract anti-elastase effect) to the introduction to motivate the study.
- Line 397: Add citation to the corresponding Table 1 for elastase results.
- Line 412: Also cite the relevant Table 3 (synergistic effect results).
- Line 420-421: Add supporting reference for phytosterols’ skin benefits (anti-aging, elasticity).
- Line 446: When referring to improved inhibitory effects, include figure number (e.g., “as shown in Figure 3”).
- Add justification for selected concentrations of hemp oil and extracts in the in-vitro assays (why 2 mg/mL or 400 µg/mL were used).
- Line 452: Cite related graph (Figure 4) showing NO inhibition.
- Make sure all claims about other studies (lines 397-423, 429-437) have proper references.
Author Response
Reply to Reviewer 2
Comments and Suggestions for Authors
The study is well-designed and provides valuable insight into the chemical and biological differences between Thai and foreign hemp seed extracts. Combining GC-MS profiling with machine learning and biological assays is innovative and relevant for cosmeceutical applications. The results are solid; however, some minor clarifications and additions are needed to strengthen the manuscript.
- The title is informative but long. You may consider a shorter version, but not mandatory.
Response: No change.
Abstract
- Avoid abbreviations (HCA, PCA, PLS-DA) before defining them.
Response: Revised. Lines 21-24.
- Add 1-2 lines introducing the project purpose e.g. “This study aimed to distinguish Thai and foreign hemp seed varieties and assess their anti-aging bioactivity through combined chemometric and biological approaches.”
Response: Revised. Lines 98-100.
- Emphasize the novelty and purpose more clearly.
Response: Revised. Lines 32-36.
Introduction
- Line 61-63: Add reference for CBD/THC reducing senescence.
Response: Reference 7.
- Line 67-68: Add reference for CBD’s effect on collagen breakdown via PI3K/Akt pathway.
Response: Reference 8.
- Line 72-75: Add citation for extraction methods (supercritical, ultrasonic).
Response: Reference 11.
- Line 75-76: Add reference for collagenase inhibition.
Response: Reference 11.
- Line 76-77: Clarify, which compounds showed the inhibition, are currently ambiguous.
Response: Revised. Lines 80-82.
- Line 96: Briefly explain the purpose of studying two enzymes (elastase and tyrosinase) in relation to aging and pigmentation.
Response: Revised. Lines 104-107.
Materials and Methods
- Line 121: Explain what “LPS-induced NO production” means.
Response: Revised. Lines 117-118.
- Line 160: Define “LPS” (lipopolysaccharide) at first mention.
Response: Revised. Line 118.
- Add justification for the selected extract concentrations.
Response: In a prior investigation, we found that hemp seed extracts inhibited microorganisms at doses between 0.128 and 2.048 mg/mL. In addition, it was discovered that the four extracts exhibited more than 80% inhibition of the alpha glucosidase, which is comparable to the positive control. Thus, a preliminary test was carried out at a concentration of 2 mg/mL for this investigation of elastase. We screened at low concentrations for the tyrosinase inhibitory test (20 µg/mL) since we need to determine the incredibly active extract for future in vivo testing. A high dose could be toxic to the animal model. As same as, the researchers selected a concentration of 400 µg/mL for the anti-inflammatory activity test to identify an extract that might demonstrate anti-inflammatory activity at a low concentration while being non-toxic to cells.
- Line 171: Avoid repeating previously defined abbreviations.
Response: Revised.
- Line 90 vs. 200: Ensure consistent use of “PCA” (Principal Component Analysis) everywhere, not “Partial Component Analysis”.
Response: Revised. Principal Component Analysis, Line 23.
- Add statistical details (number of replicates, test used, significance/confidence level).
Response: Revised. Section 2.6
- Provide rationale for choosing the 10% GC-MS abundance cutoff.
Response: Revised. The authors provided the rationale for introducing a 10% cutoff in section 2.2 of the materials and methods section.
Results & discussion
- Line 306-309: Verify figure numbering for single and combined docking images (A-F).
Response: Revised. The authors have checked and corrected the figure numbering as suggested.
- Expand explanation of the docking and redocking validation process, briefly mention why redocking is performed (to ensure the docking setup can reproduce the native ligand pose) and explain docking score interpretation (more negative = stronger binding).
Response: Revised. Section 3.6.2
- Line 393: Move a short summary of this finding (hemp extract anti-elastase effect) to the introduction to motivate the study.
Response: Since it was recently discovered that our hemp seed extract inhibits the elastase enzyme, the authors agree that this finding shouldn't be moved to the introduction.
- Line 397: Add citation to the corresponding Table 1 for elastase results.
Response: Revised. Line 479.
- Line 412: Also cite the relevant Table 3 (synergistic effect results).
Response: Revised. Line 494.
- Line 420-421: Add supporting reference for phytosterols’ skin benefits (anti-aging, elasticity).
Response: Line 505, References 39, 40
- Line 446: When referring to improved inhibitory effects, including figure number (e.g., “as shown in Figure 3”).
Response: Revised. Line 528.
- Add justification for selected concentrations of hemp oil and extracts in the in-vitro assays (why 2 mg/mL or 400 µg/mL were used).
Response: In a prior investigation, we found that hemp seed extracts inhibited microorganisms at doses between 0.128 and 2.048 mg/mL. In addition, it was discovered that the four extracts exhibited more than 80% of the alpha glucosidase, which is comparable to the positive control. Thus, a preliminary test was carried out at a concentration of 2 mg/mL for this investigation of elastase and tyrosinase activity.
As a guideline for future cellular investigations of hemp seed extracts, the researchers selected a concentration of 400 µg/mL for the anti-inflammatory activity test to identify an extract that might demonstrate anti-inflammatory activity at a low concentration while being non-toxic to cells.
- Line 452: Cite related graph (Figure 4) showing NO inhibition.
Response: Revised. Line 537.
- Make sure all claims about other studies (lines 397-423, 429-437) have proper references.
Response: Yes.
Thank you very much,
Best regards,
Authors
17.10.2025

Reviewer 3 Report
Comments and Suggestions for Authors
The manuscript presents an investigation into the anti-aging potential of hemp seed extracts, combining metabolomic re-analysis with biological validation and molecular docking. However, several methodological aspects require clarification or improvement to strengthen the scientific rigor and reproducibility of the study:
Please clarify the relationship with the previous publication. The current manuscript heavily relies on GC–MS data already published in your previous study. While the re-analysis and new biological targets are relevant, it is essential to clearly state that this is a secondary analysis of previously published data. The novelty should be explicitly framed around the new biological assays (elastase, NO, tyrosinase), the synergistic evaluation of selected metabolites, and the discriminative modeling between cultivars.
As in your previous work, compound identification was based only on spectral matching with NIST and Wiley libraries. No confirmation with standards was performed, which limits the confidence in compound annotation. Additionally, quantification is based on relative abundance without internal standards or calibration curves. This should be acknowledged as a limitation, especially since biological conclusions are drawn from specific metabolite levels. Given the objectives of the work, I believe this is a significant limitation.
The use of HCA, PCA, and PLS-DA is appropriate, but the rationale for applying a 10% abundance cutoff to the GC–MS data is not sufficiently justified. This threshold may exclude low-abundance but biologically relevant compounds. Moreover, the PLS-DA model shows moderate predictive power (Q² = 0.3733), which should be discussed more critically in terms of generalizability and robustness.
The biological validation of elastase inhibition and NO suppression is well described and replicated. However, dose–response curves and IC₅₀ values would provide more informative insights into potency. The synergistic effects between vitamin E, clionasterol, and linoleic acid are okay, but the experimental design (e.g., concentrations used, statistical analysis) should be described in more detail.
Comments on the Quality of English Language
The manuscript would benefit from careful language editing to improve clarity and readability. Several sentences are overly long or complex, which may hinder comprehension.
Author Response
Reply to Reviewer 3
Comments and Suggestions for Authors
The manuscript presents an investigation into the anti-aging potential of hemp seed extracts, combining metabolomic re-analysis with biological validation and molecular docking. However, several methodological aspects require clarification or improvement to strengthen the scientific rigor and reproducibility of the study:
- Please clarify the relationship with the previous publication. The current manuscript heavily relies on GC–MS data already published in your previous study. While the re-analysis and new biological targets are relevant, it is essential to clearly state that this is a secondary analysis of previously published data. The novelty should be explicitly framed around the new biological assays (elastase, NO, tyrosinase), the synergistic evaluation of selected metabolites, and the discriminative modeling between cultivars.
Response: Since hemp is a plant with a wide range of uses and properties, this study has taken previously presented data on the composition of hemp seed extracts and improved upon it by reanalyzing the data to compare and evaluate the anti-aging activity of two aging-related enzymes, tyrosinase and elastase, as well as the anti-inflammatory activity of hemp seed extracts from Thai and foreign strains.
- As in your previous work, compound identification was based only on spectral matching with NIST and Wiley libraries. No confirmation with standards was performed, which limits the confidence in compound annotation. Additionally, quantification is based on relative abundance without internal standards or calibration curves. This should be acknowledged as a limitation, especially since biological conclusions are drawn from specific metabolite levels. Given the objectives of the work, I believe this is a significant limitation.
Response: I think adding a limitation section or paragraph mentioning our limitations would be beneficial.
- The use of HCA, PCA, and PLS-DA is appropriate, but the rationale for applying a 10% abundance cutoff to the GC–MS data is not sufficiently justified. This threshold may exclude low-abundance but biologically relevant compounds. Moreover, the PLS-DA model shows moderate predictive power (Q² = 0.3733), which should be discussed more critically in terms of generalizability and robustness.
Response: The authors explained why the 10% relative abundance cutoff was introduced section 2.2.1). The authors also emphasize the PLS-DA analysis limitation model regarding generalization and robustness (lines 522-529).
- The biological validation of elastase inhibition and NO suppression is well described and replicated. However, dose–response curves and IC₅₀ values would provide more informative insights into potency. The synergistic effects between vitamin E, clionasterol, and linoleic acid are okay, but the experimental design (e.g., concentrations used, statistical analysis) should be described in more detail.
Response: I'm grateful for your guidance. In future investigations, the IC50 values and dose-response curves will be examined. Section 2.3 describes them for additional experiments.
Comments on the Quality of English Language
The manuscript would benefit from careful language editing to improve clarity and readability. Several sentences are overly long or complex, which may hinder comprehension.
Response: Thank you, we asked Dr. Saffanah Mohd AbAzid, English Specialist, Faculty of Pharmaceutical Sciences, Prince of Songkla University to proof our revised manuscript already, I hope this will improve the quality of our paper.
Thank you very much,
Best regards,
Authors
17.10.2025

Reviewer 4 Report
Comments and Suggestions for Authors
The manuscript presents consistent data and an innovative approach. With improvements in clarity, conciseness, explicit discussion of limitations, and minor English/style adjustments, it meets the necessary conditions of scientific content and rigor for publication in the journal.
The minor corrections/revisions are required.

Minor English/style adjustments.
Author Response
Reply to Reviewer 4
Comments and Suggestions for Authors
The manuscript presents consistent data and an innovative approach. With improvements in clarity, conciseness, explicit discussion of limitations, and minor English/style adjustments, it meets the necessary conditions of scientific content and rigor for publication in the journal.
The minor corrections/revisions are required.
Comments on the Quality of English Language
Minor English/style adjustments.
Submission Date
19 September 2025
Date of this review
02 Oct 2025 17:38:58
From pdf file (peer-review-50748267.v1.pdf):
Specific comments
Keywords
Line 39: Please change “….PLS-DA, Molecular docking, Synergistic effect” to “ … PLS-DA; Molecular docking; Synergistic effect”
Response: Revised. Lines 41-42.
- Introduction
Lines 46-47: Consider revising “… all of which are combined to make a variety of goods” to “… all of which are used to produce a wide range of products.” The revised version more clearly conveys that each component is utilized in different applications.
Response: Revised. Lines 49-50.
Line 53-55: This passage - “Hemp oil has an impressive lipid composition, including polyunsaturated fatty acids like α-linolenic acid (omega-3) and linoleic acid (omega-6). γ-linolenic acid (GLA) and stearidonic acid, two important fatty acids, are abundant in hemp seed oil.” - is a bit awkward because the first sentence says “including polyunsaturated fatty acids such as …” and then it adds “γ-linolenic acid and stearidonic acid”, which are also polyunsaturated fatty acids. To make it more natural and fluent, please consider restructuring it into a single sentence.
Response: Revised. Lines 56-59.
Lines 59-77: The paragraph is dense and covers multiple topics (phytocannabinoids, wound healing, fatty acids, oil properties, extraction residues) without breaks. While this is acceptable, the transitions between topics are sometimes abrupt and reduces readability. Some terms could be clarified for precision; for example, “increasing cell proliferation” might be more formally expressed as “promoting cell proliferation” in scientific writing. Minor wording adjustments and splitting information into separate paragraphs could further improve clarity without compromising cohesion.
Response: Revised. Line 67.
Line 78: Consider changing “In the past, the cultivation of hemp began in China around 2700 BC” to “Hemp cultivation began in China around 2700 BC.” which is more concise and typical for scientific writing.
Response: Revised. Line 83.
Line 90: “… , and partial component analysis (PCA)”. The standard statistical term is “principal component analysis (PCA)”, as referred to in line 199 of the manuscript.
Response: Revised. Line 95.
Lines 92-94: Consider the following revised version of the sentence: “Therefore, the authors aim to re-analyze their previous GC-MS metabolomic fingerprints of Thai and foreign cultivars using these algorithms (HCA, PLS-DA, and PCA).” This is fully formal and suitable for publication.
Response: Revised. Lines 100-102.
- Materials and Methods
Line 105-121: The last two sentences of the paragraph seen somewhat disconnected from the methods-focused description that precedes them. The preceding sentences describe sample preparation and extraction, then suddenly the text shifts to the study’s objectives. To improve the flow, the objectives could be moved to the beginning of the paragraph: “In this study, we focused on examining variations in chemical composition among four different hemp seed extracts, which may lead to differing capabilities in inhibiting the aging enzyme. We also assessed the effectiveness of all extracts on LPS-induced NO production and cell survival. Four hemp seed samples were categorized into two groups…”
Response: Revised. Lines 115-118.
Lines 109-110: Please alter “All samples were extracted to obtain oil, ethanol extract, and hexane extract that were provided in a previous study” to “All samples were extracted to obtain oil, ethanol, and hexane extracts, as reported in a previous study”
Response: Revised. Lines 123-126.
Line 130: The abbreviations R² (coefficient of determination) and Q² (predictive ability parameter) appear here for the first time in the manuscript. Each abbreviation should be written out in full at its first occurrence before using the abbreviated form throughout the rest of the text. I recommend that the authors define R² and Q² explicitly here.
Response: Revised. Lines 156-57.
Lines 136-137: In the sentence “Then, the solution of 1.6 mM N-Succinyl-Ala-Ala-Ala-p-nitroanilide were added.”, the verb should be singular because the subject “solution” is singular. Therefore, it should read: “Then, the solution of 1.6 mM N-Succinyl-Ala-Ala-Ala-p-nitroanilide was added.”
Response: Revised. Line 181.
Line 142: The sentence introducing equation (1) currently ends without any punctuation. Since it directly precedes the equation, it should end with a colon. Suggested revision: “… which was calculated using the following equation (1):”
Response: Revised. Line 186.
Line 147: The citation number ([18]) should follow the author(s)’ name(s), rather than appearing immediately after ‘by’ without any author reference. The authors should either include the author names before the citation number or remove ‘by’ and adjust the sentence to fit the numbered style.
Response: Revised. Line 194.
Line 153: Please alter “… calculated by the following equation (1).” to “… calculated according to equation (1).”
Response: Revised. Line 200.
Lines 159-161: The original sentence contained a few issues that hinder its understanding. A revised version for clarity is as follows: “RAW 264.7 cells were seeded at 8 × 10⁵ cells/mL in 24-well plates and activated by incubation in medium containing LPS (1 μg/mL) in the presence of various concentrations of test compounds dissolved in dimethyl sulfoxide (DMSO).”
Response: Revised. Lines 206-208.
Line 162: “... secret ...” is awkward and uncommon. Please change to “secreted” or “produced”
Response: Revised. Line 209.
Line 180: Please change “[15] and [20]” to “[15,20]”
Response: Revised. Line 232.
Lines 185-187: A single, well-structured sentence could link these two sentences/actions more smoothly, reduce redundancy, and improve readability. Consider connecting the two steps in a way that highlights the workflow logically—preparation followed by simulation—without repeating the phrase “the authors used.”
Response: Revised. Lines 237-239.
- Results
Lines 199-200: Here, it is no longer necessary to spell out the full term, as the abbreviations have already been defined in the Introduction section, and can be used directly.
Response: Revised. Line 251.
Lines 217-218: There’s a small issue in your sentence: linoleic acid and ethyl ester are listed twice, and the punctuation around the list is a bit confusing. Please rewrite.
Response: Revised. Line 281.
Line 234: Change “… extracts, Figure 2B.” to “… extracts (Figure 2B).”
Response: Revised. Line 286.
Line 267: Alter “… (4.91% - 17.63%).” to “… “… (4.91 - 17.63%).“
Response: Revised. Line 335.
Line 287: Alter “… 26% - 29%, …” to “… “… 26 - 29%, …”
Response: Revised. Line 353.
Line 301: Please change “… file, Figure S4.” to “… file (Figure S4).”
Response: Revised. Line 372.
Line 339: Table 3 should be moved to the next page so that it appears entirely on a single page.
Response: Revised. Table 4.
Line 353: Remove the percent symbol (%) after the number 70.
Response: Revised. Line 432.
- Discussion
Lines 369 and 375: As mentioned earlier (lines 199–299), it is sufficient to use the abbreviation without needing to spell out the full term again.
Response: Revised. Lines 448, 454.
Lines 375-377: Consider changing the sentence to “However, when PCA, an unsupervised learning method, was applied, it revealed that the variation among the thirteen metabolic traits was not the dominant factor driving the differences in classification.” in order to improve readability and grammatical flow.
Response: Revised. Lines 454-456.
Line 382: Please alter “In conclusion, even the differences between thirteen… “ to “In conclusion, although the differences among the thirteen …”
Response: Revised and deleted.
Line 384: Change “… were not dominant (PCA failed), ...” to “were not dominant (as PCA indicated), ...”
Response: Revised and deleted.
Lines 399-401: The two sentences — “Nevertheless, at the designated doses, these extracts exhibited no tyrosinase inhibitory activity. The tyrosinase inhibitory activity was not related to the fatty acids of the hemp seed extracts.” – can be combined into a single, well-structured sentence to improve cohesion, fluency, and readability, while maintaining the logical flow of results. Please adjust.
Response: Revised. Lines 483-485.
Line 403: Remove the percent symbol (%) after 11.1.
Response: Revised. Lines 486-487.
Lines 407-408: The sentence: “As a result, to prevent melanin pigment in hemp seed extracts in the future, the extraction method must be modified or improved.” can be confusing and misleading. The extracts don’t contain melanin; they act on melanin production in biological systems. A clearer phrasing would connect the idea back to biological inhibition and the need for methodological refinement. Please revise accordingly.
Response: Revised. Lines 490-491.
Lines 413: Alter “... elastase synergistically ...” to ““... elastase was synergistically ...”
Response: Revised. Line 496.
Line 415: The abbreviation IC₅₀ appears in the text without first being written out in full or accompanied by an explanation of its meaning. Please revise accordingly.
Response: Revised. Line 498.
Line 416: Replace “Linoleic acid” with “It” at the beginning of the sentence to minimize repetitions.
Response: Revised. Line 500.
Line 423: Throughout the manuscript clionasterol and γ-sitosterol are referred to as the same compound. That being said, the plural phrasing (“… they are suitable …”) is inconsistent. Please verify.
Response: Revised. Line 506.
Lines 427-429: You should either list all items with “and” at the end without “etc.”, or use “etc.” without “and” to indicate continuation.
Response: Revised. Line 512.
References
Line 516: In the first bibliographic reference, the number “1” appears duplicated.
Response: Deleted. Line 601.
Thank you very much,
Best regards,
Authors
17.10.2025

Reviewer 5 Report
Comments and Suggestions for Authors
This study reanalyzed the gas chromatography-mass spectrometry data of previously published four cannabis seed varieties using machine learning algorithms (Hierarchical Cluster Analysis, Principal Component Analysis, and Partial Least Squares-Discriminant Analysis). The research successfully established a novel discriminant model to distinguish between Thai and foreign cannabis seed extracts, confirming that vitamin E, sterols, and linoleic acid are significant metabolic markers. The study found that the extracts possess potential bioactivity without causing cytotoxicity. The model developed in this research contributes to further advancing product standardization and sustainability. Here are some specific comments.
- As an important foundation for this article, the author needs to further elaborate on the previous research findings in the preface.
- The data in the abstract should be accurate and specific.
- Abbreviations in the abstract should be accompanied by their full names when first appearing.
- Add a "Statistical Analysis" section to elaborate on the data statistics and significance testing in the manuscript.
- Further refine the statement of the method.
- The figures have poor readability, especially Figure 1, S1, and S2. The quality of both figures and tables in the supplementary materials is inadequate.
- The data in the manuscript need to be subjected to significance testing.
- In the discussion, it is also necessary to further strengthen the comparison with previously published research and highlight the innovation of reanalysis.
Author Response
Reply to Reviewer 5
Comments and Suggestions for Authors
This study reanalyzed the gas chromatography-mass spectrometry data of previously published four cannabis seed varieties using machine learning algorithms (Hierarchical Cluster Analysis, Principal Component Analysis, and Partial Least Squares-Discriminant Analysis). The research successfully established a novel discriminant model to distinguish between Thai and foreign cannabis seed extracts, confirming that vitamin E, sterols, and linoleic acid are significant metabolic markers. The study found that the extracts possess potential bioactivity without causing cytotoxicity. The model developed in this research contributes to further advancing product standardization and sustainability. Here are some specific comments.
- As an important foundation for this article, the author needs to further elaborate on the previous research findings in the preface.
Response: Revised. Lines 91-92.
- The data in the abstract should be accurate and specific.
Response: Revised. Lines 27-30.
- Abbreviations in the abstract should be accompanied by their full names when first appearing.
Response: Revised. Lines 21-24.
- Add a "Statistical Analysis" section to elaborate on the data statistics and significance testing in the manuscript.
Response: The authors added the statistical analysis section and elaborated on statistical and significant testing in sections 2.2.2 and 2.6
- Further refine the statement of the method.
Response: Revised. Sections 2.1, 2.2, 2.2.1, 2.2.2, 2.3 and 2.6.
- The figures have poor readability, especially Figure 1, S1, and S2. The quality of both figures and tables in the supplementary materials is inadequate.
Response: The authors provide new figures with better readability.
- The data in the manuscript needs to be subjected to significance testing.
Response: Revised.
- In the discussion, it is also necessary to further strengthen the comparison with previously published research and highlight the innovation of reanalysis.
Response: Revised. Lines 530-532.
Thank you very much,
Best regards,
Authors
17.10.2025

Round 2
Reviewer 3 Report
Comments and Suggestions for Authors
The authors revised the manuscript in agreement with the reviewers and highlighted any limitations.
Some minor comments:
In the abstract, the term “gas chromatography–mass spectroscopy (GC–MS)” should be corrected to “gas chromatography/mass spectrometry (GC/MS). Spectroscopy is another technique.
Lines 92, 127, 137, please remove the period before the reference.
The use of a 10% relative abundance cutoff in GC/MS data should briefly justify this threshold with a reference or rationale.
Some figures (e.g., Figures 1 and 2) would benefit from more detailed legends, especially to clarify color coding and sample groupings. This would improve readability and interpretation for the reader.
The manuscript would benefit from a language editing to improve clarity and readability.
Author Response
Reply to Reviewer 3 Round -2
Thank you very much for your suggestions, we tried our best to correct as you suggest as:
- In the abstract, the term “gas chromatography–mass spectroscopy (GC–MS)” should be corrected to “gas chromatography/mass spectrometry (GC/MS). Spectroscopy is another technique.
Response: Revised.
- Lines 92, 127, 137, please remove the period before the reference.
Response: Revised.
- The use of a 10% relative abundance cutoff in GC/MS data should briefly justify this threshold with a reference or rationale.
Response: The authors explained why we introduced a 10% threshold in Lines 254 -263.
“ … First, the authors analyzed the original GC/MS data, which included sixty-one metabolic features from all twelve samples using HCA, PCA, and PLS-DA analyses. However, we could not observe a distinct pattern from the original GC/MS data via HCA and PCA analyses (Figures S1 – S3, supplementary file). On the other hand, the PLS-DA analysis was able to differentiate Thai from foreign cultivars with a high R2 value of 0.9850, indicating high model fit, yet the predictive power of the PLS-DA model was in a negative value, Q2 = -0.7665, showing a low predictability of the model (Figure S4, supplementary file). Therefore, the authors introduce a 10% GC/MS relative abundance cutoff threshold to eliminate noise signals from minor chemical metabolic feature influences for a better analysis. …”
The authors further explain why 10% was selected. In Line 264 -266.
“… The authors selected 10% threshold criterion following a previous report by Santajit and team, considering any metabolite with ³ 10% relative abundance as a major component [25]. …”
- Some figures (e.g., Figures 1 and 2) would benefit from more detailed legends, especially to clarify color coding and sample groupings. This would improve readability and interpretation for the reader.
Response: The authors revised figures (Figures 1 to 3), adding more details on axis labels and legends to increase readability and interpretation for readers.
Best regards,
Authors
21.10.2025

Reviewer 5 Report
Comments and Suggestions for Authors
The author has responded to the reviewers' comments and revised the manuscript accordingly.
Author Response
Reply to Reviewer 5-Round 2
Thank you very much for your suggestion.
Best regards,
Authors,
21.10.2025